# Using Satellite Imagery to Improve Local Pollution Models for High-Voltage Transmission Lines and Insulators

Peter Krammer [1], Marcel Kvassay [1], Ján Mojžiš [1], Martin Kenyeres [1], Miloš Očkay [1,*], Ladislav Hluchý [1], Ľuboš Pavlov [2] and Ľuboš Skurčák [2]

[1] Institute of Informatics, Slovak Academy of Sciences, Dúbravská Cesta 9, 845 07 Bratislava, Slovakia; peter.krammer@savba.sk (P.K.); marcel.kvassay@savba.sk (M.K.); jan.mojzis@savba.sk (J.M.); martin.kenyeres@savba.sk (M.K.); ladislav.hluchy@savba.sk (L.H.)

[2] VUJE, a.s., 918 64 Trnava, Slovakia; lubos.pavlov@vuje.sk (Ľ.P.); lubos.skurcak@vuje.sk (Ľ.S.)

* Correspondence: milos.ockay@savba.sk; Tel.: +421-960-423-031

**Abstract:** This paper addresses the regression modeling of local environmental pollution levels for electric power industry needs, which is fundamental for the proper design and maintenance of high-voltage transmission lines and insulators in order to prevent various hazards, such as accidental flashovers due to pollution and the resultant power outages. The primary goal of our study was to increase the precision of regression models for this application area by exploiting additional input attributes extracted from satellite imagery and adjusting the modeling methodology. Given that thousands of different attributes can be extracted from satellite images, of which only a few are likely to contain useful information, we also explored suitable feature selection procedures. We show that a suitable combination of attribute selection methods (relief, FSRF-Test, and forward selection), regression models (random forest models and M5P regression trees), and modeling methodology (estimating field-measured values of target variables rather than their upper bounds) can significantly increase the total modeling accuracy, measured by the correlation between the estimated and the true values of target variables. Specifically, the accuracies of our regression models dramatically rose from 0.12–0.23 to 0.40–0.64, while their relative absolute errors were conversely reduced (e.g., from 1.04 to 0.764 for the best model).

**Keywords:** regression modeling; electric power industry; satellite images; attribute selection; local pollution modeling; stochastic variable; model refinement





## 1. Introduction

Over the past few decades, artificial intelligence in the form of machine learning and deep learning has penetrated almost all areas of human activity. These techniques have found application in such seemingly disparate areas as industry, education, science, transportation, sports, or entertainment. Typically, it is advantageous to use deep learning whenever there are sufficient quantities of training data available. Nevertheless, one can still find application areas (or cases) where big training data simply are not available or would be too costly to provide. In such cases, conventional machine learning methods may be preferred, at least until sufficient quantities of data are collected for the application of deep learning.

One such application area is the electric power industry, where the designers of new high-voltage transmission lines are required to take into account local environmental pollution levels, which are affected by many factors, such as local orography and meteorology, agricultural activity, transportation, factories, construction sites, etc. Measurement of local pollution levels in line with the established power industry standards and procedures (e.g., STN 33 0405) can easily take one year or longer; it comprises a series of field measurements of pollution deposits (each lasting six to eight weeks) followed by time-consuming laboratory analyses. Collecting big training data in this way would require simultaneous field

measurements in thousands of different locations over several years, which is currently unfeasible. As a result, application of deep learning methods in this area is significantly constrained, and classical machine learning methods—those able to cope with smaller training sets—seem to be more promising.

In general, the existing literature on insulator pollution can be divided into several groups. The first contains studies such as [1–5], which try to relate the occurrence of flashovers to the shape of the insulator, its level of pollution, and the chemical composition of the deposits. The second group looks for suitable and safe techniques to determine the degree of pollution of insulators, sometimes even during their operation under high voltage. Thus, Ferreira et al. [6] proposed and validated a method for determining the degree of pollution by spectral analysis of acoustic emissions in close proximity of polluted insulators. Maraaba et al. [7] analyzed images of pollution using a digital camera in HSV (hue, saturation, value) space. Wang et al. [8] proposed a noncontact method based on spectral analysis of plasma formed from settled contamination after the application of a short laser pulse (LIBS—Laser Induced Breakdown Spectroscopy) and Jin et al. [9] used an information fusion of infrared, visible, and ultraviolet spectrum. Other researchers rely on hyperspectral imaging and microwave radiation theory.

The third group tries to estimate the quantity and chemical composition of dust deposits that adhere to a given type of insulator in a given time, weather, location, and other key factors. The International Council on Large Electric Systems (CIGRE) recommends five ways of quantifying this type of pollution: by equivalent salt deposit density (ESDD), surface conductivity, leakage current, pollution flashover voltage (PFV), or pollution flashover gradient. Zhenghao He et al. [10] have established that high-voltage insulators are polluted both by natural sources and by human activity. Their findings indicated that metal cations mainly originate from human activity and accumulate on the bottom side of the insulators. Chen and Zhang [11] proposed a dynamic model estimating the non-soluble deposit density (NSDD) based on selected meteorological parameters. Qiao et al. [12] proposed a similar model estimating the equivalent salt deposit density (ESDD) based on the so-called grey system theory. The experimental work of Ferma et al. [13] showed a strong correlation between the amount of deposits and the atmospheric concentrations of dust particles for different European locations, so long as they were far from large sources of pollution. The same research suggested that seasonal fluctuations in the concentration of air pollutants were much higher in urban areas than in rural ones.

The vast majority of deposit and pollution models in the cited literature are only partially theoretically motivated and chiefly rely on empirical studies. The main reason is that these are complex scientific problems with many variable nonlinear factors, which have so far defied a complete theoretical explanation. Artificial intelligence is currently the most effective way of studying and solving such problems.

Our own work in this area started with the intuition that the levels of environmental pollution as defined by STN 33 0405 (on the basis of pollution deposits trapped in deposit collectors) should correlate well with standard indicators of atmospheric pollution, such as PM10, already measured on a regular basis by the Slovak Hydrometeorological Institute (SHMU) [14]. We hypothesized that with sufficient collected data, it would be possible to train reasonably accurate machine learning models to be capable of estimating the levels of pollution deposits from the atmospheric pollution indicators provided by the SHMU. This could potentially save a lot of effort associated with time-consuming field measurements according to STN 33 0405. Our initial attempts in this direction [15], however, ended in failure. Neither nonlinear transformations of atmospheric pollution indicators, nor their joint combinations have shown significant correlations with the levels of pollution deposits from our field measurements. Analysis of these results and their possible causes, which we briefly reiterate in the following sections, led us to search for new sources of information, such as satellite imagery, which we expected would be more tightly related to the levels of pollution deposits measured in the field. We also felt the need to redefine the target attributes and directly estimate the field-measured levels of pollution deposits, rather than

their upper bounds, as we had done earlier. As the results presented below demonstrate, the combination of satellite imagery with this new methodology significantly increased the accuracy of models.

The rest of this article is structured as follows: Section 2 describes potential improvement directions that we considered before embarking on the course of action reported in this paper. Section 3 describes the input and output data, and also lists possible reasons for our earlier modeling failure. Section 4 provides a brief outline of the new approach. Section 5 details the process of attribute selection and modeling, and also reports the achieved estimation results. Section 6 contextualizes these results and evaluates their statistical significance. Finally, Section 7 summarizes our findings and outlines directions for future work.

## 2. Potential Model Improvement Directions

In spite of the above-mentioned problems and our unsuccessful early attempts to model the upper bounds of target variables with sufficient precision for real-world applications, we identified several directions that might lead to higher modeling accuracy in the future, namely:

1. A significant increase in the number (spatial density) of stations for field measurements of pollution deposits, along with their long-term operation so as to obtain longer time series for a more detailed analysis;
2. A change in approach: to shift from modeling the upper bound estimates of pollution deposits to modeling their individual six-week field-measured values, from which their upper bound estimates could be subsequently calculated by appropriate statistical methods;
3. To add new types of relevant input attributes that are typically ignored by domain experts in the fields of electric power and meteorology, but which could still capture significant events affecting local pollution levels in the monitored locations;
4. To develop new statistical models of true probability distributions underlying the input and output (target) variables. Analysis of these distributions, their deviations from the expected distribution, and their detailed examination could help in identifying useful properties for their successful modeling and perhaps also reveal the reasons why our previous models failed to achieve the desired accuracy.

Subsequent elaboration of these improvement directions revealed numerous issues and challenges.

### 2.1. Increasing the Number of Field Measurements

A radical increase in the number of measuring stations and their denser spatial distribution is problematic due to the need for long-term deployment of numerous specialized measuring devices, which might be costly and require long-term investments. Nevertheless, the need to develop accurate estimation models is now pressing, forcing us to look for alternative approaches and cheaper sources of relevant information.

### 2.2. Modeling Field-Measured Values of Pollution Deposits Rather Than Their Upper Bounds

We consider it advantageous to shift from modeling the upper bound estimates of target variables to direct modeling of their field-measured values, since, for many machine learning techniques, the latter is a more natural and feasible task. Therefore, it might be both easier and more accurate to directly model their field-measured values, and then estimate their upper bounds by standard statistical methods.

This methodological shift also allows us to more precisely capture seasonal variation: During a given six-week field measurement, it is reasonable to assume that seasonal factors stay roughly constant, an assumption that clearly does not hold for the global yearly cycle. As a result, individual deviations or anomalies of each variable in these six-week periods are captured more precisely, which is a prerequisite for higher modeling accuracy.

### 2.3. Define and Collect New Relevant Input Attributes

Additional relevant input attributes may be the key to increased modeling accuracy. They can be defined in different ways, come from different sources, and be different in their physical nature, e.g., laboratory analyses of pollution deposit samples, database of main polluters such as the National Emission Information System (NEIS) [16], recordings of acoustic emissions of polluted insulators, visual information extracted from satellite images, etc. Laboratory analysis of collected pollution deposit samples is the standard and established method, but is also very time-consuming and costly. Therefore, cheaper and simpler alternatives of comparable quality are very much needed. One extremely useful source of data that comes virtually free is satellite imagery. Since we already possess quite extensive satellite imagery from Sentinel-2 satellites, we chose to use it for this purpose. Our main motivation is that abrupt color changes in the vicinity of our field measurement locations should correlate well with agricultural and other invasive interventions, such as plowing, sowing, building, or surface mining activities, which tend to increase local pollution levels.

### 2.4. Determine True Probability Distributions of Our Input and Output (Target) Variables

In general, the application of statistical approaches that take into account true statistical distributions of both input and output variables is rather rare in machine learning. The electric power industry standard STN 33 0405 states that the levels and relevant properties of trapped pollution deposits should conform to the three-parameter Weibull distribution. Although this cannot be ruled out at present, our preliminary experiments indicate that other probability distributions might describe them even more precisely. We lack the necessary number of samples to settle this issue now, but in the near future we expect to obtain atmospheric pollution indicators with much finer granularity in time (on a daily or even hourly basis), which should permit much more confident estimates of their true underlying probability distributions. We expect that this would lead to a deeper understanding of this difficult problem area and, eventually, to more reliable estimates of local pollution deposits themselves. At present, however, we are forced to postpone this line of research and exploration.

Having considered all of the above-mentioned issues and constraints, we decided to proceed with a combination of the options elaborated on in Sections 2.2 and 2.3. In other words, we tried to directly estimate the field-measured values of pollution deposits rather than their upper bounds on the basis of new input attributes primarily derived from satellite imagery.

### 3. Available Data

For this study, a measurement campaign lasting from June 2020 to May 2021 and consisting of 400 six-week field measurements in 50 different geographical locations around Slovakia (see Figure 1) provided the values of three target variables that together characterize the local environmental pollution levels (see Appendix A):

S—total pollution deposit
Sr—soluble fraction of pollution deposit
g02—electrical conductivity of soluble fraction

The campaign produced eight six-week measurements per variable per location. The established procedure as per STN 33 0405 mandates fitting a Weibull distribution to the eight values of each variable obtained at each location, and then estimating its upper bound as the 99.5-th percentile of the fitted distribution. It means that, in a given location, there is only a 0.5% chance that the variable will exceed its local upper bound. Based on these three local upper bounds (for S, Sr, and g02), each location is assigned to one of four grades of environmental pollution (I–IV) and, for each grade, appropriate parameters of insulation and maintenance procedures are prescribed.

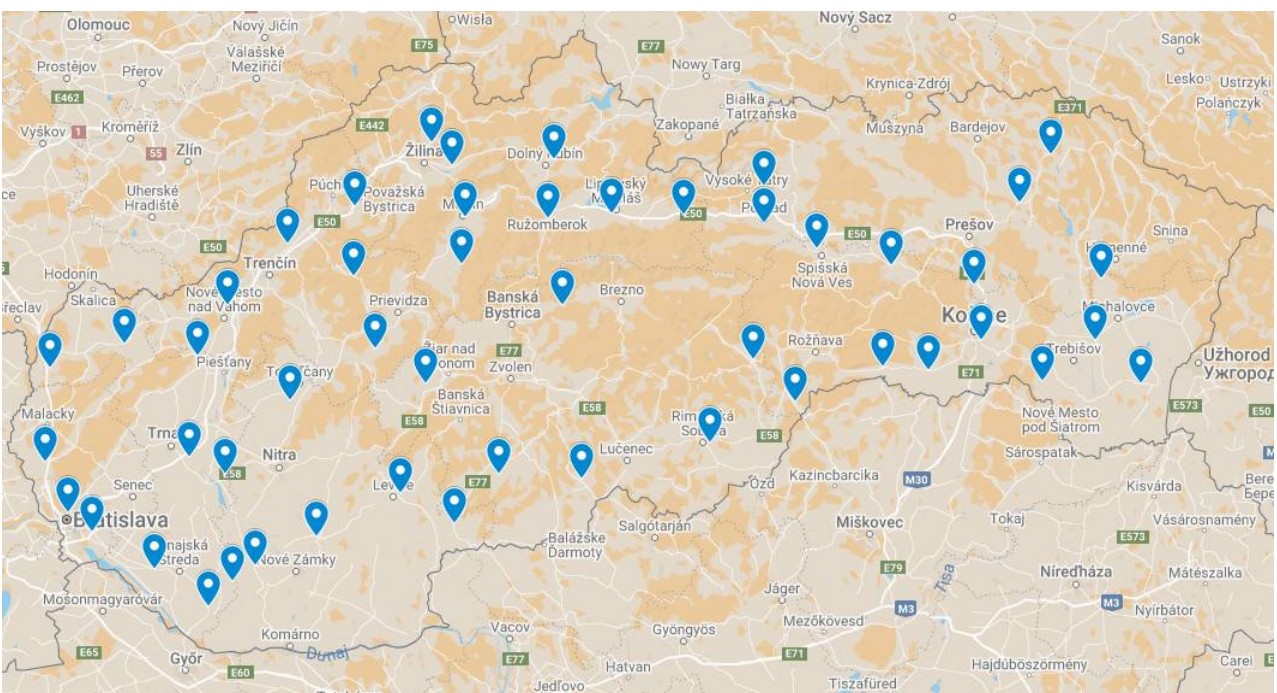

**Figure 1.** Locations of field measurements in our 2020–2021 campaign around Slovakia.

We also had access to standard atmospheric pollution indicators regularly monitored and evaluated by the Slovak Hydrometeorological Institute (SHMU) [14], which consist of the yearly average of atmospheric concentrations of dust particles and selected gasses over Slovakia. These are the same attributes that we had used in our previous work [15], and we refer to them here as the SHMU input attributes. They comprise:

- PM2.5—yearly average of concentrations of dust particles with a diameter less than 2.5 μm;
- PM10—yearly average of concentrations of dust particles with a diameter less than 10 μm;
- $O_3$—yearly average of ozone concentrations;
- $NO_2$—yearly average of nitrogen dioxide concentrations;
- $SO_2$—yearly average of sulfur dioxide concentrations.

In our previous work [15], we carried out several experiments in an attempt to reliably estimate the upper bounds of target variables S, Sr, and g02 on the basis of SHMU input attributes, but we failed to achieve reliable estimates. Neither nonlinear transformations of these attributes, nor their joint combinations, resulted in a substantial increase in the accuracy of our models. Possible reasons for this failure include:

- An insufficient number of data records (i.e., insufficiently representative data set), which does not allow for generalizing the relationship between the inputs and the outputs;
- Missing input variables with dominant effect on target attributes, which makes the behavior of target variables appear to be random;
- Unknown confidence intervals for input and output attributes that reduced the accuracy of the used models;
- An insufficient spatial density of the measured input and output values;
- An inherent difficulty of learning to reliably estimate upper bounds of random variables.

Overall, this seemed to indicate that the problem was much more difficult than it originally appeared, and a fundamental change in approach along with new relevant input attributes would be necessary to tackle it successfully. Therefore, this study exploits new input attributes primarily derived from satellite imagery.

The collection of Sentinel-2 satellite images used for this purpose aligned with the timing of our field measurements. Each image covered an area of 100 km × 100 km and its maximum spatial resolution was 10 m per pixel. (Spatial resolution actually depends on the sampled spectral band—see Table 1). Since the lift-off of the second Sentinel-2B satellite in 2017, temporal resolution of these images is one image per five days [17]. These images can be either Level 1 (capturing the top of atmosphere reflectances) or Level 2 (bottom of atmosphere reflectances). We used the B10 spectral band (cirrus cloud detection) from Level 1 images, and B1, B2, B3, B4, B6, B8, B11, and AOT spectral bands from Level 2 images [18]. We also used several normalized difference indices to indicate the prevalence of bare soils (NDSI), flourishing vegetation (NDVI), water (NDWI), and moisture (NDMI) in a given pixel [19]. (An overview of these spectral bands is provided in Table 1).

**Table 1.** List of used satellite spectral bands including normalized difference indices.

| Name | Scale | Pixel Size | Wavelength | Description |
|---|---|---|---|---|
| AOT | 0.001 | 10 m | | Aerosol optical thickness |
| B1 | 0.0001 | 10 m | 443.9 nm | Aerosols |
| B2 | 0.0001 | 10 m | 496.6 nm | Blue |
| B3 | 0.0001 | 10 m | 560.0 nm | Green |
| B4 | 0.0001 | 10 m | 664.5 nm | Red |
| B6 | 0.0001 | 20 m | 740.2 nm | Red Edge 2 |
| B8 | 0.0001 | 10 m | 835.1 nm | NIR |
| L1 B10 cir | 0.0010 | 60 m | 1373.5 nm | Cirrus |
| B11 | 0.0001 | 20 m | 1613.7 nm | SWIR 1 |
| NDVI (normalized difference vegetation index) | 0.0001 | 10 m | | NDVI = (B8 − B4)/(B8 + B4) |
| NDWI (normalized difference water index) | 0.0001 | 10 m | | NDWI = (B3 − B8)/(B3 + B8) |
| NDSI (normalized difference soil index) | 0.0001 | 20 m | | NDSI = (B3 − B11)/(B3 + B11) |
| Moisture index | 0.0001 | 20 m | | moisture index = (B8 − B11)/(B8 + B11) |

Since the low number of our field measurements (data records) did not permit the application of deep learning (which could directly process raw images), we had to construct the satellite input attributes manually. This first exploratory experiment considers the reflectance values of pixels no further away than 500 m from each of the fifty chosen measurement locations. Each image pixel in such a circular measurement area was represented by one time series of reflectance values for each spectral band captured by the satellite in each six-week field-measurement period. Given that we used 13 spectral bands (including the derived ones), a total of 13 time series of basic reflectance values for each considered pixel and each six-week measurement period was obtained. Construction of one such time series for one pixel is symbolically shown in Figure 2.

Figure 2 indicates that basic reflectance values themselves may not suffice—their differences may actually carry more useful information. Therefore, from each basic reflectance series, several derived ones were created. The first of them was the simple difference series, which contained the difference of two consecutive values in the original reflectance series. Since these differences can be negative, we also constructed an absolute difference series, which only contained positive values. Then, a normalized difference series was created, where the same difference was divided by the number of days separating the two consecutive satellite photos. Finally, we also constructed an absolute value version of the

normalized difference series, termed absolute normalized difference. In this way, five time series for each spectral band and each pixel in the measurement area were created. Given that there were 13 spectral bands, this process led to $5 \times 13 = 65$ time series per pixel.

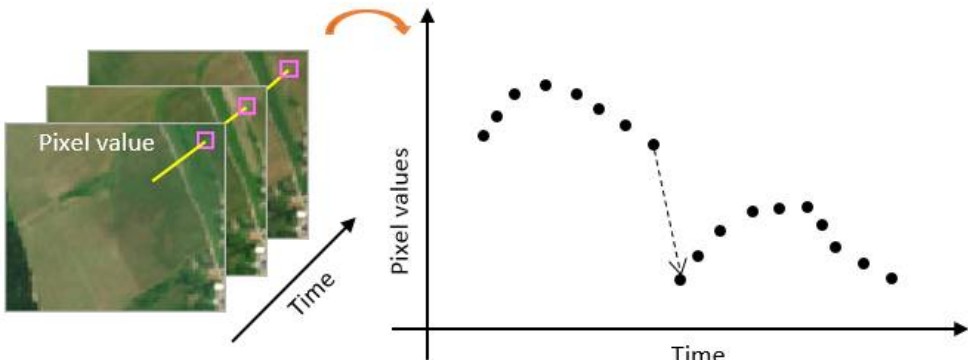

**Figure 2.** (**Left**) Series of Sentinel 2 satellite photos of the same geographical area with one chosen pixel in the top right corner. (**Right**) Symbolic depiction of the time series of the reflectance values in one spectral band for the chosen pixel, with an abrupt change in the middle (indicated by the dashed arrow), which might signify an invasive human activity, such as plowing or harvesting.

Considering the high number of pixels in each measurement area, we decided to aggregate these time series by six basic statistical functions: average, minimum, maximum, and three main quantiles (25%, 50%, and 75%). In order to get rid of individual pixels, we aggregated the values twice: first along the time axis for each pixel separately, and then spatially, for all pixels in a given measurement area. Since all possible combinations of the six basic statistical functions were permitted, this produced $6 \times 6 = 36$ unique double aggregation combinations. In other words, we calculated not only spatial average of time averages, but also spatial average of time medians, minima and maxima, spatial minimum of time averages and medians, as well as of time minima and maxima, etc. Overall, this resulted in $36 \times 65 = 2340$ numbers, which captured (in a very condensed form) the overall dynamics of various processes in each measurement area during one six-week measurement period. These 2340 numbers were then passed on to the machine learning models as the satellite input attributes.

## 4. Brief Outline of Our Approach

In general, our approach conformed to the CRISP-DM model, a de facto industry standard process for data mining, schematically depicted in Figure 3.

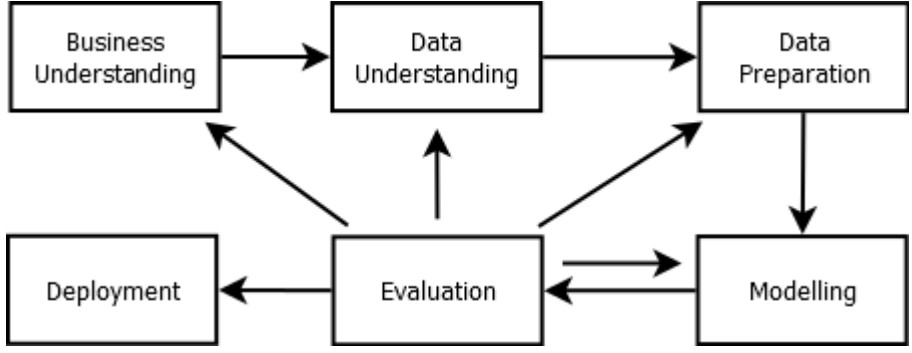

**Figure 3.** Schematic depiction of data mining according to CRISP-DM.

Our earlier research reported in [15] represents one iteration of the CRISP-DM cycle, in which the target attributes were the long-term upper bounds of variables S, Sr, and g02, and the input attributes were the SHMU input attributes described in Section 3. Since

this iteration was not successful, the work described here can be taken to represent the second iteration of the same CRISP-DM process, initiated by the return from the Evaluation phase back to the Data Understanding phase, in which satellite imagery was identified as a promising source of new and more informative input attributes, and the methodology was changed to estimating the field-measured values of target variables rather than their long-term upper bounds.

The construction of satellite attributes described in the previous section, then, comprises the first part of the Data Preparation phase, which ends with the attribute selection described in the next section. The choice of suitable regression models and their subsequent training jointly represent the Modelling phase. Finally, Section 6 (Discussion) and Section 7 (Conclusions) present the contextualized evaluation of the results.

The methodological shift to modelling the field-measured values of target variables rather than their long-term upper bounds helps to capture their seasonal variation. This raises an interesting question: how much of that variation is stable, i.e., could be reliably predicted purely on the basis of time, even without satellite imagery? In order to answer that question, we decided to pass to machine learning models the date of each six-week field measurement. In the text, we call it the measurement date, but in fact it is the Day of Year (1–365) referring to the day in the middle of the concerned six-week measurement period. We would like to point out that this attribute does not uniquely identify individual field measurements, because it has the same value for all 50 measurement locations in a given six-week period (since all of the field measurements were recorded synchronously). Therefore, there is no danger of overfitting regression models on the basis of this new attribute.

For the sake of completeness, we note that we also passed to machine learning models the five SHMU input attributes, as well as geographical coordinates of each measurement location (latitude, longitude). Thus, the total number of input attributes rose to 2348. Our general hypothesis was that the attributes derived from satellite imagery would help to significantly improve the accuracy of models.

In order to evaluate their accuracy, we combined n-fold cross validation with standard regression criteria, namely: the correlation coefficient, the relative absolute error, and the root relative squared error. Due to the limited number of available records, we used 40-fold cross validation, which ensured sufficient stability of the estimated model accuracy. One of the most serious drawbacks of this kind of validation is that it takes a lot of time, since the training process has to be repeated 40 times. On the other hand, resultant estimates represent the accuracy of trained models precisely, stably, and objectively.

## 5. Selection of Attributes, Modeling, and Results

Due to the large number of new attributes derived from satellite data, attribute selection, and dimensionality, reduction of the input attribute space was needed. Prior to this process, all input attributes were normalized to a uniform range: interval <0, 1>. This ensured that all input attributes were equally treated, since some attribute selection methods are sensitive to this aspect.

In the first round of experiments, neither the attribute selection based on correlation coefficient (in which the attributes showing the highest absolute value of Pearson or Spearman correlation with the target attribute were used), nor dimension reduction on the basis of principal component analysis (PCA) ensured satisfactory results. Models trained as such actually achieved lower or similar accuracy as before. This failure stemmed from the fact that high correlation with the target attribute can indicate the impact of a given input attribute, but only individually, in isolation from others, because their mutual correlation is ignored. Therefore, combining such input attributes need not increase the resultant model accuracy at all, if they are strongly correlated. On the other hand, the failure of PCA stemmed from the opposite reason: it is an unsupervised method, and can give no guarantee that its main components would be strongly correlated with the target attribute.

For these reasons, we had to select input attributes differently, by applying methods that took into account their mutual correlations and could correctly gauge their joint effect

on the target attribute. After some preliminary trials, we chose the relief method [20], FSRF-Test [21], and forward selection [22], whose applications resulted in significant improvements in the accuracy of trained models. Selected attributes were incrementally added in groups of five; therefore, the resultant models used 5, 10, 15, or 20 input attributes. We illustrate these steps in Figure 4, which represents a more detailed elaboration of the Data Preparation, Modeling, and Evaluation phases of the CRISP-DM model depicted in Figure 3.

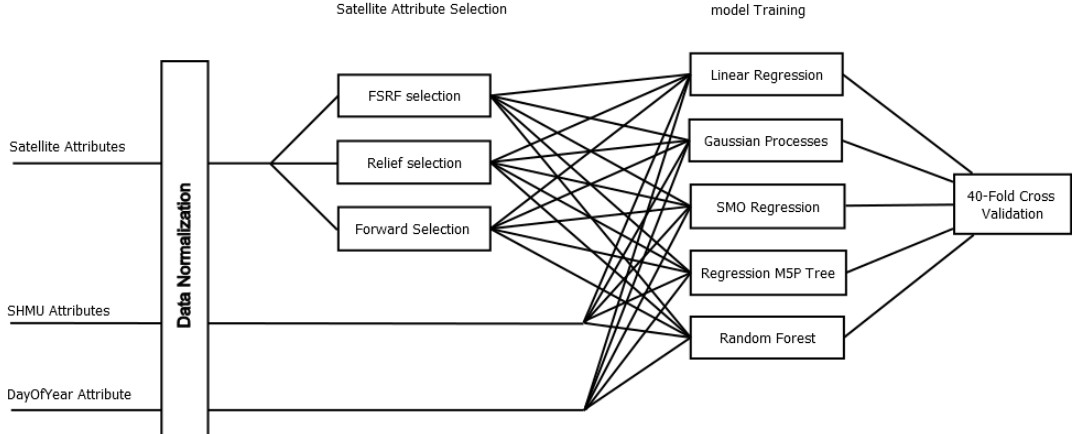

**Figure 4.** Flowchart view of our data preparation, modeling, and evaluation activities.

In the following Tables 2–4, we present the best results that we achieved for the three target attributes S, Sr, and g02, respectively, in terms of the correlation coefficient between their true and predicted field-measured values, their relative absolute error and their root relative squared error.

**Table 2.** Overview of trained regression models and their accuracy metrics for the field-measured values of target variable S (total deposit).

| | Linear Model | Gaussian Process | SMO Regression | M5P Tree | Random Forest |
|---|---|---|---|---|---|
| Models using attributes SHMU, without Day of Year and without Satellites | | | | | |
| Correlation coefficient | 0.2195 | 0.2289 | 0.2377 | 0.1921 | 0.2243 |
| Relative absolute error | 0.960381 | 0.958122 | 0.838752 | 0.963788 | 0.996284 |
| Root relative sq. error | 0.975826 | 0.971225 | 1.013767 | 0.983825 | 1.007389 |
| Models using attributes SHMU and Day of Year, without Satellites | | | | | |
| Correlation coefficient | 0.3044 | 0.3228 | 0.3221 | 0.3655 | 0.3810 |
| Relative absolute error | 0.953682 | 0.931039 | 0.81304 | 0.873916 | 0.867675 |
| Root relative sq. error | 0.952256 | 0.944247 | 0.990007 | 0.933188 | 0.960127 |
| Models using attributes SHMU, Satellites (FSRF-TEST 20 attr), without Day of Year | | | | | |
| Correlation coefficient | 0.3962 | 0.3962 | 0.3936 | 0.4011 | 0.5015 |
| Relative absolute error | 0.929776 | 0.91984 | 0.776143 | 0.913888 | 0.834525 |
| Root relative sq. error | 0.918874 | 0.916169 | 0.971942 | 0.928474 | 0.863463 |
| Model using attributes SHMU, Satellites (Relief 20 attr), without Day of Year | | | | | |
| Correlation coefficient | 0.2934 | 0.3322 | 0.2697 | 0.3291 | 0.3574 |
| Relative absolute error | 0.998795 | 0.958923 | 0.827651 | 0.952494 | 0.930419 |
| Root relative sq. error | 0.969389 | 0.945013 | 1.009543 | 0.972802 | 0.936803 |
| Models using attributes SHMU, Satellites (Forward selection 20 attr), without Day of Year | | | | | |
| Correlation coefficient | 0.3754 | 0.3691 | 0.2771 | 0.2891 | 0.4214 |
| Relative absolute error | 0.977038 | 0.942754 | 0.808266 | 1.005264 | 0.851436 |
| Root relative sq. error | 0.934184 | 0.928854 | 0.999298 | 1.010215 | 0.905755 |

**Table 2.** *Cont.*

|  | Linear Model | Gaussian Process | SMO Regression | M5P Tree | Random Forest |
|---|---|---|---|---|---|
| Models using attributes SHMU, Satellites (Forward selection 10 attr), Day of Year | | | | | |
| Correlation coefficient | 0.4288 | 0.4475 | 0.4104 | 0.4263 | 0.3976 |
| Relative absolute error | 0.948856 | 0.89844 | 0.778747 | 0.922759 | 0.861133 |
| Root relative sq. error | 0.911549 | 0.892596 | 0.962801 | 0.916577 | 0.921097 |
| Models using attributes SHMU, Satellites (Forward selection 5 attr), Day of Year | | | | | |
| Correlation coefficient | 0.4635 | 0.4508 | 0.4506 | 0.3769 | 0.4381 |
| Relative absolute error | 0.923401 | 0.896762 | 0.762388 | 0.924544 | 0.843924 |
| Root relative sq. error | 0.886452 | 0.890677 | 0.952388 | 0.948177 | 0.898847 |
| Models using attributes SHMU, Satellites (FSRF-TEST 10 attr), Day of Year | | | | | |
| Correlation coefficient | 0.4363 | 0.4339 | 0.4242 | 0.4686 | 0.5400 |
| Relative absolute error | 0.900255 | 0.892416 | 0.761173 | 0.868294 | 0.806506 |
| Root relative sq. error | 0.899348 | 0.899034 | 0.956712 | 0.885832 | 0.84258 |
| Models using attributes SHMU, Satellites (FSRF-TEST 5 attr), Day of Year | | | | | |
| Correlation coefficient | 0.3956 | 0.3884 | 0.3983 | 0.4255 | 0.5183 |
| Relative absolute error | 0.925902 | 0.911766 | 0.768899 | 0.885788 | 0.805674 |
| Root relative sq. error | 0.918372 | 0.919311 | 0.969704 | 0.905563 | 0.853435 |
| Models using attributes SHMU, Satellites (Relief 10 attr), Day of Year | | | | | |
| Correlation coefficient | 0.4016 | 0.4063 | 0.3959 | 0.4021 | 0.3987 |
| Relative absolute error | 0.934508 | 0.910194 | 0.781673 | 0.908593 | 0.863821 |
| Root relative sq. error | 0.921802 | 0.913254 | 0.960806 | 0.921916 | 0.916560 |
| Models using attributes SHMU, Satellites (Relief 5 attr), Day of Year | | | | | |
| Correlation coefficient | 0.3533 | 0.3364 | 0.3490 | 0.3028 | 0.3347 |
| Relative absolute error | 0.922774 | 0.915231 | 0.784270 | 0.912179 | 0.904339 |
| Root relative sq. error | 0.938938 | 0.941382 | 0.975032 | 0.972598 | 0.956348 |
|  | Linear model | Gaussian process | SMO Regression | M5P Tree | Random forest |

**Table 3.** Overview of trained regression models and their accuracy metrics for the field-measured values of target variable Sr (soluble fraction of deposit).

|  | Linear Model | Gaussian Process | SMO Regression | M5P Tree | Random Forest |
|---|---|---|---|---|---|
| Models using attributes SHMU, without Day of Year, and without Satellites | | | | | |
| Correlation coefficient | 0.2302 | 0.1822 | 0.1934 | 0.2314 | 0.1898 |
| Relative absolute error | 0.961795 | 0.974067 | 0.855066 | 0.960243 | 1.006067 |
| Root relative sq. error | 0.971534 | 0.981626 | 1.014425 | 0.971155 | 1.016997 |
| Models using attributes SHMU, Day of Year without Satellites | | | | | |
| Correlation coefficient | 0.2582 | 0.2236 | 0.2305 | 0.2053 | 0.2643 |
| Relative absolute error | 0.952205 | 0.962065 | 0.857284 | 0.962442 | 0.913962 |
| Root relative sq. error | 0.964646 | 0.973101 | 1.006680 | 0.988054 | 1.005748 |
| Models using attributes SHMU, Satellites (FSRF-TEST 20 attr) without Day of Year | | | | | |
| Correlation coefficient | 0.1428 | 0.1786 | 0.1742 | 0.065 | 0.3264 |
| Relative absolute error | 0.991643 | 0.970063 | 0.859727 | 1.034066 | 0.920992 |
| Root relative sq. error | 0.998483 | 0.984765 | 1.012168 | 1.097823 | 0.945923 |
| Models using attributes SHMU, Satellites (Relief 20 attr) without Day of Year | | | | | |
| Correlation coefficient | 0.0927 | 0.0648 | 0.1264 | 0.0789 | 0.1826 |
| Relative absolute error | 1.016839 | 1.016156 | 0.877836 | 1.064591 | 0.967896 |
| Root relative sq. error | 1.014047 | 1.007755 | 1.023097 | 1.049356 | 1.005119 |

**Table 3.** *Cont.*

|  | **Linear Model** | **Gaussian Process** | **SMO Regression** | **M5P Tree** | **Random Forest** |
|---|---|---|---|---|---|
| Models using attributes SHMU, Satellites (Forward selection 20 attr) without Day of Year | | | | | |
| Correlation coefficient | 0.1883 | 0.1210 | 0.233 | 0.1469 | 0.1377 |
| Relative absolute error | 1.005077 | 0.986239 | 0.835902 | 1.031335 | 0.953701 |
| Root relative sq. error | 0.995576 | 1.000356 | 0.992946 | 1.015140 | 1.007062 |
| Models using attributes SHMU, Satellites (Forward selection 10 attr), Day of Year | | | | | |
| Correlation coefficient | 0.2398 | 0.3101 | 0.3535 | 0.1445 | 0.2839 |
| Relative absolute error | 1.022736 | 0.948921 | 0.815373 | 1.054828 | 0.870082 |
| Root relative sq. error | 0.995421 | 0.951980 | 0.966335 | 1.135082 | 0.961922 |
| Models using attributes SHMU, Satellites (Forward selection 15 attr), Day of Year | | | | | |
| Correlation coefficient | 0.2611 | 0.3029 | 0.3283 | 0.2056 | 0.2714 |
| Relative absolute error | 1.007581 | 0.959974 | 0.805193 | 0.998895 | 0.878738 |
| Root relative sq. error | 0.982944 | 0.955482 | 0.972581 | 1.021172 | 0.962601 |
| Models using attributes SHMU, Satellites (FSRF-TEST 10 attr), Day of Year | | | | | |
| Correlation coefficient | 0.2888 | 0.323 | 0.3305 | 0.3005 | 0.3698 |
| Relative absolute error | 0.947149 | 0.919711 | 0.804691 | 0.937511 | 0.866946 |
| Root relative sq. error | 0.964184 | 0.945773 | 0.969808 | 0.960605 | 0.928711 |
| Models using attributes SHMU, Satellites (FSRF-TEST 5 attr), Day of Year | | | | | |
| Correlation coefficient | 0.3299 | 0.3438 | 0.36 | 0.3284 | 0.3999 |
| Relative absolute error | 0.936913 | 0.913589 | 0.793238 | 0.938661 | 0.866407 |
| Root relative sq. error | 0.945296 | 0.937339 | 0.965524 | 0.945354 | 0.921331 |
| Models using attributes SHMU, Satellites (Relief 10 attr), Day of Year | | | | | |
| Correlation coefficient | 0.2194 | 0.2357 | 0.2848 | 0.2272 | 0.3102 |
| Relative absolute error | 0.964405 | 0.956218 | 0.846194 | 0.963997 | 0.869070 |
| Root relative sq. error | 0.986047 | 0.975347 | 0.987452 | 0.988660 | 0.949752 |
| Models using attributes SHMU, Satellites (Relief 5 attr), Day of Year | | | | | |
| Correlation coefficient | 0.2587 | 0.2837 | 0.309 | 0.2601 | 0.2705 |
| Relative absolute error | 0.957589 | 0.94709 | 0.826091 | 0.961476 | 0.897960 |
| Root relative sq. error | 0.970585 | 0.958345 | 0.981613 | 0.971070 | 0.971982 |
|  | Linear model | Gaussian process | SMO Regression | M5P Tree | Random forest |

**Table 4.** Overview of trained regression models and their accuracy metrics for the field-measured values of target variable g02 (electrical conductivity).

|  | **Linear Model** | **Gaussian Process** | **SMO Regression** | **M5P Tree** | **Random Forest** |
|---|---|---|---|---|---|
| Models using attributes SHMU, without Day of Year, and without Satellites | | | | | |
| Correlation coefficient | −0.0888 | 0.006 | −0.0239 | 0.076 | 0.1191 |
| Relative absolute error | 1.013206 | 1.011356 | 0.996436 | 1.000262 | 1.041810 |
| Root relative sq. error | 1.014285 | 1.005392 | 1.018875 | 1.000869 | 1.044912 |
| Models using attributes SHMU, Day of Year without Satellites | | | | | |
| Correlation coefficient | 0.4113 | 0.4373 | 0.4342 | 0.6206 | 0.6339 |
| Relative absolute error | 0.903762 | 0.894292 | 0.886614 | 0.755594 | 0.756144 |
| Root relative sq. error | 0.911274 | 0.897203 | 0.905929 | 0.782397 | 0.778181 |
| Modeling using attributes SHMU, Satellites (FSRF-TEST 20 attr), without Day of Year | | | | | |
| Correlation coefficient | 0.4195 | 0.4314 | 0.405 | 0.3931 | 0.4062 |
| Relative absolute error | 0.904127 | 0.892004 | 0.914068 | 0.91234 | 0.910343 |
| Root relative sq. error | 0.908831 | 0.899933 | 0.919583 | 0.925257 | 0.924097 |

**Table 4.** *Cont.*

| | Linear Model | Gaussian Process | SMO Regression | M5P Tree | Random Forest |
|---|---|---|---|---|---|
| Models using attributes SHMU, Satellites (Relief 20 attr), without Day of Year | | | | | |
| Correlation coefficient | 0.4135 | 0.3942 | 0.3798 | 0.3853 | 0.4601 |
| Relative absolute error | 0.903282 | 0.905045 | 0.910703 | 0.912017 | 0.881088 |
| Root relative sq. error | 0.912632 | 0.918536 | 0.935326 | 0.929368 | 0.889698 |
| Models using attributes SHMU, Satellites (Forward selection 20 attr), without Day of Year | | | | | |
| Correlation coefficient | 0.3155 | 0.3276 | 0.3257 | 0.3426 | 0.4708 |
| Relative absolute error | 0.942383 | 0.931576 | 0.946136 | 0.910879 | 0.873131 |
| Root relative sq. error | 0.953646 | 0.946278 | 0.955216 | 0.951316 | 0.883867 |
| Models using attributes SHMU, Satellites (Manual selection 3 attr), Day of Year | | | | | |
| Correlation coefficient | 0.5163 | 0.5245 | 0.5092 | 0.6275 | 0.6004 |
| Relative absolute error | 0.855031 | 0.848659 | 0.849584 | 0.76101 | 0.78007 |
| Root relative sq. error | 0.855314 | 0.849386 | 0.863360 | 0.77716 | 0.800759 |
| Models using attributes SHMU, Satellites (Forward selection 5 attr), Day of Year | | | | | |
| Correlation coefficient | 0.5562 | 0.5565 | 0.5305 | 0.6058 | 0.6434 |
| Relative absolute error | 0.827117 | 0.82688 | 0.835880 | 0.788339 | 0.761934 |
| Root relative sq. error | 0.830543 | 0.829219 | 0.852279 | 0.795252 | 0.764349 |
| Models using attributes SHMU, Satellites (FSRF-TEST 5 attr), Day of Year | | | | | |
| Correlation coefficient | 0.5635 | 0.5661 | 0.5354 | 0.5608 | 0.626 |
| Relative absolute error | 0.822849 | 0.82399 | 0.850483 | 0.825195 | 0.762599 |
| Root relative sq. error | 0.825459 | 0.822494 | 0.849352 | 0.830468 | 0.779014 |
| Models using attributes SHMU, Satellites (Relief 5 attr), Day of Year | | | | | |
| Correlation coefficient | 0.4891 | 0.5099 | 0.4858 | 0.5494 | 0.6368 |
| Relative absolute error | 0.87403 | 0.861349 | 0.869334 | 0.822957 | 0.759978 |
| Root relative sq. error | 0.872218 | 0.858429 | 0.881012 | 0.837942 | 0.769692 |
| | Linear model | Gaussian process | SMO Regression | M5P Tree | Random forest |

We compared several different model types and noted that the most encouraging results were typically achieved by linear regression, gaussian processes, SMO regression (support vector machine with improved sequential minimal optimization), M5P regression tree (regression version of C4.5 with pruning and smoothing), and random forest (consisting of 100 regression trees): see the Tables 2–4 above. In most cases, the top performer was random forest. We attribute this to the fact that it internally employs ensemble learning and that particular trees in it are trained independent of each other, which effectively prevents overfitting.

## 6. Discussion

Among the attributes derived from satellite images, there was significant variability in their contribution to the accuracy of the resultant regression models. Moreover, there were significant differences in their contribution not only for different target attributes but also for different attribute selection methods.

In general, the highest accuracy was typically achieved by random forest models, which we plan to use in the near future when we obtain additional attributes, either extracted from the NEIS system or derived from acoustic emissions of polluted insulators. We also plan to use new satellite-derived attributes, since we do not think our present ones were able to extract all of the useful information from satellite images.

From the perspective of attribute selection methods, the highest model accuracies were achieved with FSRF-Test, which proved to be the most suitable in four cases (for S and Sr target attributes). For the target attribute g02, the forward selection and the relief methods achieved better results, each in one case.

As for the number of selected input attributes, there is no clear universally optimal number; it largely depends on the degree of relevance of input attributes to a given target attribute. If highly relevant input attributes are available (e.g., from NEIS [16] or from acoustic measurements), the number of required input attributes may significantly decrease, while, paradoxically, the accuracy of the resultant model may increase.

For individual target variables S, Sr, and g02, before the introduction of the new input attributes (i.e., without the measurement date and the satellite-derived ones), the correlation of their estimated values with the true ones was 0.23, 0.23, and 0.12, respectively. After adding the satellite-derived attributes, these correlations rose to 0.50, 0.33, and 0.46, respectively. The maximum correlations were achieved after the measurement date (Day of Year, number of the middle day) was also added; the correlations then rose to 0.54, 0.40, and 0.64, respectively. Among the three modeled target variables, the highest accuracy was achieved for g02, where the relative absolute error was 0.762 (down from original 1.04) and the correlation coefficient was 0.64 (up from original 0.12). This striking difference can also be observed in Figure 5: on the left are the results of our earlier attempts to estimate the long-term upper bounds of g02 purely from the SHMU input attributes [15], and on the right, the results of our present attempt to directly estimate the field-measured values of g02 from satellite-derived attributes and the measurement date. In both cases, the ideal situation in which the estimated values equal the predicted ones is represented by the dashed diagonal line, while the solid green line represents the actual linear regression function. Although it is evident that more work needs to be done before these models can actually be used in practice, we consider these results a success and an important milestone.

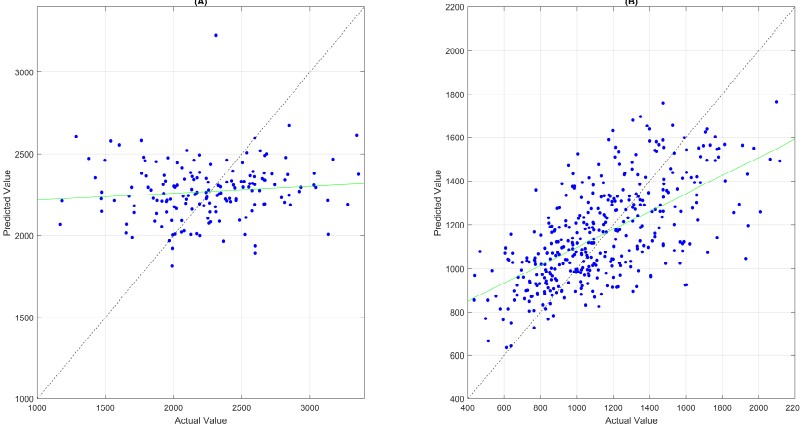

**Figure 5.** Correlation of actual values (horizontal axis) with predicted values (vertical axis) for (**A**) long-term upper bounds of g02 estimated from the SHMU input attributes, and (**B**) field-measured values of g02 estimated from the satellite-derived attributes and the measurement date.

Finally, we would like to point out that we observed significant seasonality for all target variables, which was clearly shown by their high correlations with the measurement date. In fact, the effect of the measurement date alone nearly equaled the combined effect of all our satellite-derived input attributes. This means that the local levels of environmental pollution undergo relatively stable seasonal variations of a very general character, at least as represented by the three target variables S, Sr, and g02.

*Statistical Tests*

We used statistical permutation tests in order to evaluate the statistical significance of observed increases in model accuracy after adding the new input attributes. (In these tests, the model accuracy was expressed by the Pearson correlation coefficient between the estimated and the true values of target variables.) This type of nonparametric test is used when there are two groups of samples (2-sample test), for which we want to test

whether they have identical means or medians. In other words, we wanted to be sure that the observed increases in model accuracy were real and not just random flukes.

To this end, the training and validation of random forest models were repeated 30 times with different random seeds (with seed values from 1 to 30 for pseudorandom generator initialization). In each of the 30 repetitions, the same seed was used for the creation of the random forest model and for its 40-fold cross validation. Thus, sufficient objectivity of the results was ensured. The same group of seeds (1–30) was used for models with and without the additional attributes. Vectors $X_A$ and $X_B$ represent the accuracies (Pearson correlations) of the 30 trained regression models with and without extra attributes, respectively. Statistical permutation tests were then performed with the following parameters:

Test statistic: absolute difference of mean values (abs (mean $(X_A)$ − mean $(X_B)$));
Number of permutations in the statistical test = 1,000,000;
Significance level alpha = 0.05;
Null hypothesis: $X_A$ and $X_B$ have the same mean value;
Alternative hypothesis: $X_A$ and $X_B$ do not have the same mean value.

The results of these statistical tests are shown in Table 5. In all cases, the *p*-value was lower than the predetermined alpha value of 0.05. This indicates a statistically significant increase in the accuracy of trained models after the addition of extra attributes. In one case only, for target attribute g02, the *p*-value of 0.0125 was of the same order of magnitude as (but still lower than) the predefined alpha level. We believe that, in this particular case, the general seasonal variation captured by the measurement date carried nearly the same amount of relevant information as the satellite-derived input attributes. Strictly speaking, Table 5 does not show which of the two compared models in each pair was more accurate, but as seen in previous Tables 2–4, models with more new attributes in the second column $(X_B)$ were always more accurate than the models with fewer new attributes in the first column $(X_A)$.

**Table 5.** Results of statistical permutation tests, which show that additional satellite-derived attributes and the measurement date have ensured a significant increase in the accuracy of trained regression models.

| Target attribute S: | | |
|---|---|---|
| Used attributes for $X_A$ | Used attributes for $X_B$ | *p*-value |
| SHMU | SHMU + Day of Year | 0.000238 |
| SHMU | SHMU + Satellites | 0.000179 |
| SHMU + Day of Year | SHMU + Day of Year + Satellites | 0.000617 |
| SHMU + Satellites | SHMU + Day of Year + Satellites | 0.000698 |
| Target Attribute Sr: | | |
| Used attributes for $X_A$ | Used attributes for $X_B$ | *p*-value |
| SHMU | SHMU + Day of Year | 0.000297 |
| SHMU | SHMU + Satellites | 0.000119 |
| SHMU + Day of Year | SHMU + Day of Year + Satellites | 0.000917 |
| SHMU + Satellites | SHMU + Day of Year + Satellites | 0.002740 |
| Target Attribute g02: | | |
| Used attributes for $X_A$ | Used attributes for $X_B$ | *p*-value |
| SHMU | SHMU + Day of Year | 0.000249 |
| SHMU | SHMU + Satellites | 0.000314 |
| SHMU + Day of Year | SHMU + Day of Year + Satellites | 0.012501 |
| SHMU + Satellites | SHMU + Day of Year + Satellites | 0.006714 |

Another notable fact concerns the differing strengths of correlation between the target variables on the one hand, and the measurement date versus a selected group of satellite

attributes (marked in the table as Satellites) on the other. In the case of target variable S, the measurement date provides a lower increase in model accuracy than the satellite attributes. This follows from the fact that the *p*-value for the first permutation test in Table 5 (SHMU versus SHMU + Day of Year) is higher than the one for the second permutation test (SHMU versus SHMU + Satellites): the lower *p*-value means it is less likely that the two compared models have identical means. Somewhat surprisingly, target variable g02 behaves differently: here, the measurement date provides a more significant increase in accuracy (evidenced by the lower corresponding *p*-value) than the group of satellite attributes. This means that although the target variables jointly capture various aspects of local environmental pollution, each of them has its own individual behavior and dynamics, significantly different from the other two. These differences can also be gauged from the differing accuracies of the trained models for S, Sr, and g02, listed in Tables 2–4 above.

## 7. Conclusions and Future Work

By modeling individual field-measured values of target variables S, Sr, and g02 instead of their long-term upper bounds, we were able to significantly increase the accuracy of the trained regression models. This progress was made possible by the additional input attributes that were capable of capturing seasonality and other significant local events (e.g., building or surface mining activities), in combination with suitable model types and appropriate methods of attribute selection.

The new input attributes consisted of the date of measurement along with a large number of attributes derived from multispectral satellite imagery by various double combinations of six basic statistical functions. Our experiments confirmed that these attributes indirectly capture significant events and processes affecting local environmental pollution levels.

We tested the statistical significance of the increases in model accuracy after the addition of extra attributes using a nonparametric permutation test, which confirmed that the increase was significant. The magnitude of this increase was shown to depend on the method of attribute selection and on the type of regression model as well.

We found out that the effect of adding the measurement date alone was similar in size to the joint effect of all of the new satellite-derived attributes. This means that local pollution levels exhibit some strong and relatively stable seasonal cycles. Nevertheless, the experiments also confirmed that there was independent information in the satellite-derived attributes, because the most accurate models were those combining the satellite attributes with the measurement date. We believe that this combination would also prove to be the most stable of all combinations investigated so far.

In the future, we would like to derive more input attributes from satellite images in order to extract more information from them. We also plan to exploit daily or even hourly values of the SHMU input attributes, which, by virtue of representing the variation of local pollution levels more accurately, could further improve the accuracy of the models. Another potential increase in model accuracy could come from the use of the NEIS database, which contains information on main environmental polluters—industrial producers and companies. Yet another interesting research direction is the modeling of the true statistical distributions of local pollution levels, as embodied in target variables S, Sr, and g02, which might lead to a more detailed knowledge of their behavior and, consequently, to higher estimation accuracy. It is also possible that we might discover new domain knowledge potentially useful, not only for the electric power industry, but also for ecology and related disciplines.

**Author Contributions:** Data curation, P.K., J.M., Ľ.P. and Ľ.S.; Formal analysis, P.K. and M.K. (Marcel Kvassay); Methodology, P.K. and M.K. (Marcel Kvassay); Project administration, M.K. (Marcel Kvassay) and L.H.; Resources, J.M.; Software, P.K.; Supervision, L.H.; Validation, P.K. and M.K. (Marcel Kvassay); Writing—original draft, M.K. (Martin Kenyeres) and M.O.; Writing—review & editing, P.K., M.K. (Marcel Kvassay) and M.K. (Martin Kenyeres). All authors have read and agreed to the published version of the manuscript.

**Funding:** This research was funded by the project VEGA 2/0125/20 and by the Slovak Research and Development Agency under the contract no. APVV APVV-20-0548 (ARIEN).

**Institutional Review Board Statement:** Not applicable.

**Informed Consent Statement:** Not applicable.

**Data Availability Statement:** Certain portion of our data cannot be made public due to commercial agreement limitations under which we obtained it from SHMU and VUJE companies. The publicly available part of our data can be downloaded from https://zenodo.org/record/6373788 (accessed on 26 February 2022).

**Acknowledgments:** This work was supported by the project VEGA 2/0125/20 and by the Slovak Research and Development Agency under the contract no. APVV APVV-20-0548 (ARIEN). We would also like to thank the VUJE, a.s. company (Trnava, Slovakia) for field measurement data and professional consultations, and the Slovak Hydrometeorological Institute (SHMU—Bratislava, Slovakia [14]) for atmospheric pollution data and professional consultations.

**Conflicts of Interest:** The authors declare no conflict of interest.

## Appendix A. Outline of Slovak Technical Standard STN 33 0405 (Selection of Outdoor Insulators According to the Level of Environmental Pollution)

This national standard was in use from 1990 to 2015. It was partly inspired by, but more detailed than, IEC 815 (Guide for the Selection of Insulators in Respect of Polluted Conditions, 1986). Since it was not replaced by any official successor standard, it continued to be used even after its withdrawal as the standard and established procedure for the selection of outdoor insulators for high-voltage transmission lines operating in the range of 1–750 kV in Slovakia.

As per the standard, local environmental pollution should be determined by the capture of pollution deposits into special deposit containers. These containers should be placed about two meters above ground level at transmission stations and other important landmarks. At each location, there should be two or three such containers installed. The maximum recommended period of deposit capture for one field measurement was two months (the exact duration of each measurement should be recorded for each container). At the end of the measurement, pollution deposits from all containers in the same location should be first combined, then dried out and weighed. On the basis of this dry weight, the amount of total pollution deposit (S) was to be determined. By its subsequent dissolution in distilled water its soluble fraction (Sr) as well as the electric conductivity of its 0.2% solution (g02) should also be determined.

The standard mandated 6 to 12 such field measurements at each evaluated location. From each location, we obtain a series of 6 to 12 values for each variable S, Sr, and g02. To each such series, a three-parameter Weibull distribution has to be fitted. After fitting, the upper bound of each considered variable is determined as the 99.5th percentile of its fitted Weibull distribution. These three upper bounds are then used to classify each location into one of four degrees of environmental pollution: I (least polluted) to IV (most polluted). For each pollution degree, appropriate types of insulators and maintenance procedures are specified.

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
