# Peer review of "Using Satellite Imagery to Improve Local Pollution Models for High-Voltage Transmission Lines and Insulators"

_futureinternet, doi:10.3390/fi14040099_

Round 1

Reviewer 1 Report

The topic of the paper is important for the HV network planning and maintenance with respect to the estimation of pollution in the HV transmission line networks.
The use of the machine learning  is not a standard procedure of transmission lines  planning and design. The paper deals with a new application area of machine learning methods.

Good job. More details about the time and spatial resolution of satellite images used in the analysis could be an advantage.

Reviewer 2 Report

The paper introduces a research work on improving the local pollution model of high voltage power line insulators based on satellite imaging. The results of the work could be interesting and can be helpful in the pollution determination, but the manuscript requires improvement in several points:
-The abstract requires improvement. It is more a part of the introduction than a short extract of the paper in this form.
-The sections 'Introduction' and 'Related Work' are suggested to merge into a whole Introduction section highlighting the purpose of the research. However, the research purpose description (rows 80-86) is not exact. Please improve this section and clarify the research issue solved in this work. Why is the previous work of the authors not mentioned in this section? Since this work an improvement of the previous research?
-Section 4 summarises the directions of improvement. This section is a bit lengthy and not so easy to follow. This section should be a little shorter and more specific.
-Section 5 is an introduction of the research approach. The authors describe the problems, but it is not so easy to imagine. Please insert some pictures relating to the issues.
-Section 6 is a description of the data selection from satellite data and the modelling. It is not easy to follow this section. Please insert a flowchart and other charts to show the steps of modelling.
-Results please visualize the data and the results.
-The authors refer to the STN 33 0405 standard. This is a national standard. Please introduce this standard briefly, eg. in an appendix.

Round 2

Reviewer 2 Report

The paper is significantly improved. I have no further questions or comments on the manuscript.

Author Response

Thank You.